# The Angiogenesis Inhibitor Isthmin-1 (ISM1) Is Overexpressed in Experimental Models of Glomerulopathy and Impairs the Viability of Podocytes

**DOI:** 10.3390/ijms24032723

**Published:** 2023-02-01

**Authors:** Virgilia Sahiri, Jonathan Caron, Elena Roger, Christophe Desterke, Khalil Ghachem, Inna Mohamadou, Justine Serre, Niki Prakoura, Soraya Fellahi, Sandrine Placier, Sahil Adriouch, Lu Zhang, Christos E. Chadjichristos, Christos Chatziantoniou, Hans Kristian Lorenzo, Jean-Jacques Boffa

**Affiliations:** 1Sorbonne Université, UMR_S 1155, 75006 Paris, France; 2Institut National de la Santé et de la Recherche Médicale UMR_S 1155, 75020 Paris, France; 3Université Paris-Saclay, Faculté de Médecine, 94270 Le Kremlin-Bicêtre, France; 4Université Paris Saclay, INSERM UA/09 UMR-S 935, 94800 Villejuif, France; 5Inserm UMR_S 938, Centre de Recherche Saint-Antoine, Institut Hospitalo-Universitaire de Cardio-Métabolisme et Nutrition (ICAN), Sorbonne Université, 75013 Paris, France; 6UNIROUEN, INSERM, U1234, Pathophysiology, Autoimmunity, Neuromuscular Diseases and Regenerative THERapies (PANTHER), Normandie University, 76000 Rouen, France; 7Division of Nephrology, Affiliated Hospital of Nanjing University of Chinese Medicine, Jiangsu Province Hospital of Chinese Medicine, Nanjing 210004, China; 8Department of Nephrology, Bicêtre Hospital, AP-HP, 94270 Le Kremlin-Bicêtre, France; 9Université Paris Saclay, INSERM UMR_S 1197, 94803 Villejuif, France; 10Département Néphrologie et Dialyses, Tenon Hospital, AP-HP, 75020 Paris, France

**Keywords:** glomerular diseases, proteinuria, chronic kidney disease progression, FSGS, podocytes, apoptosis

## Abstract

Focal segmental glomerulosclerosis (FSGS) is a major cause of end-stage renal disease and remains without specific treatment. To identify new events during FSGS progression, we used an experimental model of FSGS associated with nephroangiosclerosis in rats injected with L-NAME (N_ω_-nitro-L-arginine methyl ester). After transcriptomic analysis we focused our study on the role of Isthmin-1 (ISM1, an anti-angiogenic protein involved in endothelial cell apoptosis. We studied the renal expression of ISM1 in L-NAME rats and other models of proteinuria, particularly at the glomerular level. In the L-NAME model, withdrawal of the stimulus partially restored basal ISM1 levels, along with an improvement in renal function. In other four animal models of proteinuria, ISM1 was overexpressed and localized in podocytes while the renal function was degraded. Together these facts suggest that the glomerular expression of ISM1 correlates directly with the progression-recovery of the disease. Further in vitro experiments demonstrated that ISM1 co-localized with its receptors GRP78 and integrin αvβ5 on podocytes. Treatment of human podocytes with low doses of recombinant ISM1 decreased cell viability and induced caspase activation. Stronger ISM1 stimuli in podocytes dropped mitochondrial membrane potential and induced nuclear translocation of apoptosis-inducing factor (AIF). Our results suggest that ISM1 participates in the progression of glomerular diseases and promotes podocyte apoptosis in two different complementary ways: one caspase-dependent and one caspase-independent associated with mitochondrial destabilization.

## 1. Introduction

Focal segmental glomerulosclerosis (FSGS) is the histological expression of a set of pathologies characterized by marked proteinuria. It is one of the leading causes of end-stage kidney disease worldwide [1]. Although the mechanisms of proteinuria are better understood, FSGS is still lacking effective treatment, so the discovery of therapeutic targets is of utmost importance.

Podocytes are the main target cell for injury in FSGS. Together with endothelial cells, they are essential for the formation of the glomerular filtration barrier. The podocyte is particularly vulnerable due to its highly differentiated postmitotic phenotype. Structural changes involved in the progression of FSGS include alterations of the cytoskeleton, effacement of the foot of podocytes, and cell death. All this implies the alteration of the glomerular basement membrane and the leakage of nonspecific plasma proteins into the urinary space. The sources of podocyte injury are varied (genetic abnormalities, circulating factors, viral infections, medications, etc.), however the effect on podocytes is similar [1]. Additionally, some vascular injuries (e.g., due to ischemia, hypertension) can cause FSGS [2]. Thus, the appearance of proteinuria has been observed in patients undergoing antitumor treatment with antiangiogenic drugs, probably resulting from an imbalance between proangiogenic/antiangiogenic factors [3].

Isthmin-1 (ISM1) is an anti-angiogenic protein involved in the apoptosis of endothelial cells. It is a 60 kDa secreted protein that contains a centrally located thrombospondin type 1 repeat (TSR) and a C-terminal AMOP domain (adhesion-associated domain in MUC4 and other proteins) [4]. The rat gene shares a strong homology (89.52% of identity) with the human gene. In adult mammals, ISM1 is expressed in many tissues including lymphocytes, heart, lung, and kidneys [5] but its physiologic role remains poorly studied. Indeed, ISM1 has never been involved in renal physiology and/or pathophysiology. Recently, Jiang and colleagues reported that mature adipocytes secrete ISM1 and trigger an insulin-like signaling cascade [6]. In endothelial cells, secreted ISM1 potently inhibits VEGF/basic fibroblast growth factor (bFGF)-induced angiogenesis [7]. Other studies demonstrated that ISM1 acts as a vascular permeability factor in lipopolysaccharide (LPS) or hypoxia models in mouse lung [8,9]. In addition, ISM1 binds two membrane receptors, integrin αvβ5 and GRP78, triggering apoptosis in tumor and endothelial cells. Interestingly, both receptors are expressed in glomeruli and have been linked to glomerular diseases [10,11,12,13]. GRP78 is an endoplasmic reticulum-resident chaperone that maintains protein homeostasis and regulates the unfolded protein response. GRP78 is upregulated in glomerular diseases and contributes to diabetic nephropathy [11]. Inhibition of integrin αvβ5 attenuates vascular permeability and protects against renal ischemia-reperfusion injury and decreased mortality in septic mice [14,15]. 

The present study aims to elucidate whether ISM1 is involved in the mechanisms leading to glomerular damage during renal disease progression particularly focused on podocytes. For this purpose, we used several models of proteinuria including L-NAME-treated rats (N_ω_-nitro-L-arginine methyl ester), a non-specific nitric oxide (NO) synthase inhibitor, an experimental model of hypertension and secondary FSGS of vascular origin [16,17,18]. Similarly, we tested the involvement of ISM1 in other in vivo models of proteinuria, as well as in cultured podocytes.

## 2. Results

### 2.1. L-NAME Induced Nephroangiosclerosis with Focal Segmental Glomerulosclerosis

Administration of L-NAME in the drinking water caused hypertension, high proteinuria and typical lesions of nephroangiosclerosis (Figure 1a,b). The mean urine protein excretion rate (UPER) of all L-NAME-treated hypertensive rats was 2170 ± 183 mg/mmol at the time of L-NAME removal, 27-fold higher compared to the control group (80 ± 19 mg/mmol *p* < 0.001). The systolic blood pressure of the W0 group was 206 ± 9 mmHg compared to 131 ± 7 mmHg for the control animals (*p* < 0.01). L-NAME administration induced renal failure, with plasma urea nitrogen of 22 ± 5 mmol/L, compared with 5 ± 0.7 mmol/L in the control animals (*p* < 0.01). Serum creatinine was 118 ± 19 μmol/L, compared with 29 ± 3 μmol/L for control animals (*p* < 0.01; Figure 1a). As described, L-NAME induced nephroangiosclerosis with focal segmental glomerulosclerosis (FSGS) lesions when UPER was greater than 1 g/mmol [19]. As expected, the W0 group was significantly more affected than the control group when considering arteriolosclerosis, interstitial fibrosis (arrow), and tubular dilatation (Figure 1b). 

One week after interruption of L-NAME treatment (W1), UPER significantly decreased from 2170 ± 183 mg/mmol to 582 ± 123 mg/mmol (*p* < 0.001), without returning to normal levels (*p* < 0.05, W1 vs CTL). After 5 weeks without treatment (W5), proteinuria was further decreased to 301 ± 39 mg/mmol. Systolic arterial hypertension was maintained despite withdrawal of L-NAME, with W1 and W5 values of 183 ± 7 mmHg and 180 ± 8 mmHg, respectively, compared to 131 ± 7 mmHg in control animals (*p* < 0.01). Plasma urea and serum creatinine returned to normal levels after stopping L-NAME, with respectively 5.0 ± 0.4 mmol/L and 46 ± 8 μmol/L in W1 (*p* < 0.01 compared with the W0 group). Five weeks after L-NAME withdrawal, values returned to normal (Figure 1a) and renal lesions improved markedly: glomerulosclerosis, tubular dilatation, arteriosclerosis, and interstitial fibrosis scores decreased (data not shown).

### 2.2. ISM1 Gene Expression Is Associated to the Intensity of Renal Injury in L-NAME-Induced Nephropathy and Its Repair

The above series of experiments allowed the formation of four groups showing different degrees of progression and/or regression of renal disease: a control group (CTL), a group with severe glomerulonephritis (W0), a group at early phase of recovery (W1), and a group with an almost complete reversion of renal disease (W5, see Figure 2a). We performed an unbiased transcriptomic analysis in the kidneys of the four groups in order to identify genes whose expression level is associated with the phases of progression/regression of renal disease (Figure 2b). We selected ISM1 not only because of the intensity of its expression but above all because of its variability, which was correlated to the recovery of renal function. This protein had never before been reported to be involved in renal pathophysiology and, furthermore, rodent genes share strong homology with humans (89.52% identity for rats). 

We found 424 genes with at least a two-fold change in expression during the progression of the disease (CTL versus W0 group, *p* < 0.05). The 25 genes with the highest increased and decreased expression are listed in Appendix A. Although ISM1 was not among the most highly expressed genes in W0, its expression correlated well with disease progression/regression. In this model, ISM1 was expressed in the W0 group and gradually returned to normal in the W1 and W5 groups. This pattern of ISM1 expression became more evident when the study was focused on isolated glomeruli, rather than total kidney tissue (microarrays). Thus, quantitative RT-PCR assays in isolated glomeruli, revealed that the W0 and W1 groups had significantly higher ISM1 expression than the control group (Figure 2c).

### 2.3. ISM1 and Receptors Are Expressed in Podocytes of Different Models of Glomerulopathy

We analyzed the renal expression of ISM1 in rat, mouse, and human kidneys in physiological conditions. A clear expression of ISM1 was observed in renal glomeruli, more pronounced in rats than in human kidneys (Figure 3a). In rat kidney tissue, ISM1 co-localized with nephrin (Figure 3b) but not with endothelial markers such as CD31 (Figure 3c) or RECA-1, a rat pan-epithelial marker (Appendix A). These results suggest a podocytic localization of ISM1 that was confirmed by immunogold electronic microscopy (Figure 3b, right). In addition, the two known ISM1 receptors, integrin αvβ5 and GRP78, were also expressed in glomeruli under basal conditions (Appendix A).

ISM1 expression was also explored in different rodent models of glomerulopathy: L-NAME, puromycin aminonucleoside (PAN), doxorubicin in rats and diabetes (ob/ob), and LPS in mice (Figure 4). All those models developed severe proteinuria associated with net hypoalbuminemia (Appendix A). RT-qPCR experiments showed a significant overexpression of ISM1 mRNA in the glomeruli of all models studied (Figure 4a). Of note, in PAN-treated rats, mRNA levels of αvβ5 integrin and GRP78 (both described as ISM1 receptors) increased concomitantly with ISM1 (Figure 4b). Compared to controls, integrin b5 subunit mRNA was significantly overexpressed in aged diabetic mice, while no significant changes were observed in the rest of the other models (Figure 4b).

Given the presence of ISM1 receptors on glomeruli, we proceeded to explore its binding to the surface of cultured podocytes. In vitro experiences revealed that ISM1 and its receptors were expressed by cultured podocytes. Co-immunoprecipitation experiments demonstrated the interaction of soluble ISM1 to human podocytes via αvβ5 integrin and GRP78 (Figure 4c). 

### 2.4. Recombinant ISM1 Induced Podocyte Injury In Vitro

To determine the degree of damage to podocytes, we examined the effect of ISM1 on the structure of the actin cytoskeleton, as well as on cell viability. To determine the degree of damage to podocytes, we examined the effect of ISM1 on the structure of the actin cytoskeleton, as well as on cell viability. Loss of F-actin stress fibers in cultured podocytes mimics changes in podocytes in vivo that are primarily characterized by a peripheral distribution of F-actin and a delocalization of synaptopodin (SYNPO), an actin-associated protein [20]. In our study, the distribution of F-actin and SYNPO was significantly modified by the strongest stimuli at high doses of ISM1 (1 µM) and longer incubations (72 h). Under these conditions, actin adopted a subcortical distribution just beneath the plasma membrane, while the presence of F-fibers was considerably reduced, which is compatible with destabilization of the podocyte cytoskeleton (Figure 5a). 

On the other hand, the addition of recombinant ISM1 to podocytes induced a dose- and time-dependent decrease in their viability (Figure 5b). This ISM1 toxicity was significantly reduced after the addition of QVD-oph, a pancaspase inhibitor (Figure 5c). Thus, podocytes were treated with ISM1 (5 and 35 µg/mL) for 24 and 72 h in the presence or absence of QVD-oph. Under milder conditions (5 µg/mL, 24 h), QVD-oph prevented ISM1-induced mortality, suggesting that caspases are activated and responsible for this effect. However, at higher doses or longer incubations, ISM1 toxicity was only partially inhibited by QVD-oph (Figure 5b right), pointing towards an additional mechanism of caspase-independent killing. This suggests that ISM1-induced apoptosis in the podocyte operates through both caspase-dependent and caspase-independent pathways that appear to act in concert. 

### 2.5. Recombinant ISM1 Induced Podocyte Apoptosis, Mitochondrial Membrane Depolarization and Release of Pro-Apoptotic Proteins

This proapoptotic activity of ISM1 was also confirmed by TUNEL assay (Figure 6a,b), as well as by the concomitant decrease of procaspase-3 and procaspase-8 due to their cleavage and activation of both (Figure 6c,d). In addition, we analyzed ROS production or calcineurin activation, both events described as destabilizing podocyte structure and viability [20]. Their inhibition, respectively by N-acetylcysteine or FK-506, had no effect on podocyte structure or viability (data not shown).

Previous works in endothelial cells demonstrated that ISM1 can target mitochondria and trigger apoptosis [21]. Similarly, we monitored mitochondrial health using the dye JC-1, which forms aggregates based on mitochondrial membrane potential (ΔΨm). Green fluorescence indicates the monomeric form of the probe, which turns red with the formation of “J-aggregates”. Then, mitochondrial depolarization is estimated by a decrease in the red/green fluorescence intensity ratio. As shown in Figure 6e, ISM1 addition in cultured podocytes induced a significant decrease of the ΔΨm. This suggests, together with the loss of cell viability (described above), the involvement of a mitochondria-dependent apoptosis.

The release of mitochondrial intermembrane space proteins such as cytochrome c or apoptosis inducing factor (AIF) are unequivocally the central event in mitochondrial-dependent apoptosis. The mechanism by which these proteins are released into the cytosol presumably depends on the cell type and the nature of the stimuli [22]. Therefore, we examined the translocation of cytochrome c and AIF to the cytosol and nucleus. Under basal conditions, AIF and cytochrome c are distributed, exhibiting a punctate profile due to their exclusively mitochondrial localization in podocytes (Figure 6f). Upon strong stimulation of podocytes with ISM1, AIF diffused and translocated to the nucleus differently than cytochrome c (Figure 6f). These data suggest that strong ISM1 stimuli induce mitochondrial destabilization in podocytes with mitochondrial membrane permeabilization and nuclear translocation of AIF, a caspase-independent apoptosis-inducing protein.

### 2.6. Proposal Roles of Mitochondria: In Silico Analysis of ISM1 Regulation in the L-NAME Model

Whole transcriptome experiments were performed on the total kidney organ during the time course of kidney regeneration with distinct time points: week 0, week 1, and week 5. Palvidis template matching algorithm was used to find the expression profile supervised on ISM1 expression as predictor because ISM1 is known to be an anti-angiogenic actor well expressed in the cortical region of the kidney, comprising a high proportion of glomeruli. After correction of enriched genes with false discovery rate adjustment, still 283 genes were found significantly co-regulated to ISM1 expression during kidney regeneration (Appendix A). Functional enrichment performed with these co-regulated genes on the Gene Ontology Cellular Component database highlighted an important implication of the mitochondria and peroxisome cellular compartments (Figure 7A). Among this expression profile associated to ISM1 regulation during kidney relation we could highlight 48 molecules implicated in the mitochondria compartment and 13 in the peroxisome compartment: nine molecules were found shared between these two compartments (Acox1, Hmgcl, Acsl1, Slc27a2, Nudt19, Ech1, Crat, Mlycd, Ehhadh) (Figure 7B). Expression heatmap of the best 75 genes correlated to ISM1 regulation during kidney regeneration revealing a cluster of 13 genes closely regulated to ISM1 (blue cluster, Figure 7C). This blue cluster contained Gadd45g particularly known to be implicated in apoptosis and caspase related functionalities, also Neu1 which is well known in kidney and apoptosis literature, Acot4 associated to peroxisome citations, follow by Mlh3 and Mapk4 associated to apoptosis function. These results suggest that during time course kidney regeneration the main genes co-regulated with ISM1 are implicated in mitochondria and peroxisome compartments and associated to caspase especially Gadd45g, Neu1, Mapk4, ISM1, and Mlh3.

## 3. Discussion

To date, most published studies on ISM1 have focused on its role during development. ISM1 was first identified in Xenopus laevis gastrula embryos with prominent expression in the isthmus region of the brain [4], hence its name. ISM1 has been described as a target for WNT/β-catenin and NODAL signaling as well as for hematopoiesis in zebrafish development [23,24]. In mammals, using a microarray database, Valle-Rios reported the first pattern expression of ISM1 in adults, in both human and mouse tissues [5]. Contrasting with embryo, ISM1 is not expressed in the central nervous system and is strongly associated with barrier tissues, skin, mucosa, and selected lymphocyte populations. In that study, the whole kidney expression was very low. Under physiological conditions, we demonstrated that ISM1 was expressed at the mRNA and protein level in adult rodents and humans. This expression is compartmentalized and specific for podocytes, as indicated by two complementary approaches, electron microscopy and colocalization immunocytochemistry using nephrin as a podocyte marker.

Very little is known about ISM1 and its eventual physiological role. ISM1 has two identified receptors: GRP78, a protein usually activated during endoplasmic reticulum stress [12,25], and integrin αvβ5, which promotes extracellular matrix stabilization and peritubular capillary permeability [13,14]. In our study, both receptors were present in rat glomeruli under normal conditions and ISM1 co-localized with GRP78, and partially with the β5 subunit of αvβ5 integrin (Appendix A). A previous study reported that immobilized ISM1 can act as a pro-survival factor in cultured endothelial cells by promoting cell adhesion and migration [26]. However, in our study, ISM1 did not appear to be significantly expressed in glomerular endothelial cells.

In our transcriptomic study of the L-NAME model, we observed a strong variability of ISM1 expression during progression of the disease. Thus, our results showed that ISM1 expression varied according to the degree of renal damage, and was confirmed by RT-qPCR analysis of isolated glomeruli. To ensure that this association was not a peculiarity of the L-NAME model, we checked the expression of ISM1 in four additional renal disease models: doxorubicin-induced nephropathy and diabetes, PAN, and LPS. In all these models, ISM1 was overexpressed related with proteinuria and in some cases was associated with the expression of its receptors. In this sense, a previous study reported increased expression of GRP78 in podocytes after PAN treatment [10]. The fact that this expression is specifically localized in podocytes suggests that the interaction of ISM1 with its receptor could promote cell damage.

Alteration of the cytoskeleton is a common feature of podocytopathies and constitutes a classic readout for their study [20,27]. Our experiments with cultured podocytes revealed that weak ISM1 stimuli did not apparently affect the structure of the cytoskeleton. On the contrary, at high doses and prolonged incubations, ISM1 promoted the progressive loss of the fibrillar pattern of F-actin which ultimately was accumulated at the subcortical level (Figure 5a). Similarly, synaptopodine (SYNPO), an actin-associated protein, significantly lost its fibrillar distribution. This effect may be due to the dephosphorylation of SYNPO by a calcineurin-dependent mechanism and its subsequent degradation [28]. However, in our model, the inhibition of calcineurin by FK506 did not prevent this effect (not shown), so this mechanism seems to be ruled out. Alternatively, it is possible that the cytoskeleton collapses in some other way, as occurs, for example, during the execution of apoptosis [29].

Indeed, we found that ISM1 significantly decreased podocyte viability in a dose-time dependent manner (Figure 5b). At low doses, ISM1 caused impairment of podocyte viability that was abrogated by Q-VD-oph, a pan caspase inhibitor, suggesting the involvement of a caspase-dependent apoptosis process. However, upon strong ISM1 stimulation, Q-VD-oph only partially inhibited lethality in podocytes, suggesting the additional presence of a caspase-independent form of death (Figure 5c). These results were supported by the TUNEL assays, as well as by observing the disappearance of the zymogen forms of procaspase-8 and procaspase-3, suggesting their activation by cleavage. Other studies reported a proapoptotic effect of ISM1 but on endothelial cells (EC) [7,26]. In these studies, it was shown that ISM1 induced EC apoptosis through a caspase-dependent pathway, since selective inhibitors of caspase-3 and -8 prevented this effect. They also showed that activation of caspase-3 and -8 by ISM1 was dependent on its binding to αvβ5 integrin. In contrast to our study, no involvement of caspase-independent activity was shown.

Classically, two main apoptotic pathways have been described: (i) the extrinsic or death receptor pathway and (ii) the intrinsic or mitochondrial pathway. The mitochondrial membrane potential (ΔΨm) is a key indicator of mitochondrial activity, as it reflects the process of electron transport and oxidative phosphorylation, essential for ATP production [28]. The drop of ΔΨm is considered an early event in the apoptotic cascade, as it occurs before nuclear degradation. In our study we showed that ISM1 triggers the drop of ΔΨm, suggesting mitochondrial apoptosis. This event is consistent with previous studies performed on endothelial cells where the cell death was mediated via mitochondria through GRP78. The authors described two different pathways: (i) activation of p53, synthesis of proapoptotic proteins (e.g., BH3-only proteins) and subsequent mitochondrial damage or, (ii) through an endosomal pathway in which ISM1, transported by SNAP25, could directly destabilize mitochondria [21]. In our experimental setting we did not observe activation of p53 (not shown), so we are tempted to privilege a direct interaction of ISM1 with mitochondria via endosomes.

The outer mitochondrial membrane permeabilization is frequently associated with a drop of ΔΨm. Depending on the system, this phenomenon causes the release of proapoptotic proteins from the intermembrane space of the mitochondria. The two main proapoptotic components of this compartment are the cytochrome c and AIF (apoptosis inducing factor) [30]. In our study we showed that ISM1 induced the translocation of AIF from the mitochondria to the cytosol and the nucleus, contrary to cytochrome c whose distribution remained apparently unchanged. AIF is a mitochondrial oxidoreductase with a dual role in cell life/death [31]. Under normal conditions, AIF exhibited a punctate distribution in control podocytes, typically associated with its physiological confinement in mitochondria After ISM1 treatment, AIF adopted a more diffuse cytosolic profile, as well as a nuclear translocation (Figure 6f, arrows) suggesting a caspase independent apoptosis [31]. In contrast, under the same conditions, we did not observe a similar pattern for the cytochrome c. Like AIF, cytochrome c acts as an apoptotic protein, but in a caspase-dependent manner through activation of caspase-9 [32]. This different release of mitochondrial proapoptotic proteins has already been reported [33], but not very well explained. In conclusion, ISM1 acts on podocytes by activating two different death pathways (caspase-dependent and caspase-independent) that seem to act in concert.

In a more speculative basis, we analyzed in silico the role of ISM1 in our L-NAME model of glomerulopathy progression-recovery. Interestingly, we observed a relevant role of the mitochondrial and peroxisomal compartments during the period of post-injury renal regeneration (Figure 7). This would be consistent with the mitochondrial destabilization described above. The so-called “peroxisome–mitochondria connection” involves close cooperation in the cellular metabolism of lipids and reactive oxygen species [34]. This functional interrelationship has obvious implications for disease development. Although this hypothesis is highly speculative, it provides interesting information on the involvement of these organelles in the development of chronic kidney disease, a hypothesis that remains to be confirmed and explored.

In the light of our findings, we need to underline that at the glomerular level, ISM1 would be produced and secreted by podocytes. Subsequently, it would exert a proapoptotic effect on the same podocytes but also on endothelial cells, according to other previous works. This suggests that ISM1 exerts a pro-apoptotic role through autocrine (podocytes) or paracrine (endothelial cells) regulation. As a final result, the destruction of the structures that integrate the glomerular filtration barrier is ensured, proteinuria appears, and disease occurs. Our proposed model is illustrated in Figure 8.

## 4. Materials and Methods

### 4.1. Reagents and Antibodies

Recombinant ISM1 (endotoxin free) was produced by myBiosource (MBS1208159) expressed in yeast and purified near to homogeneity. Antibodies for immunofluorescence: anti-ISM1 was provided from Thermo Fischer (PA5-24968, 1:100 dilution); anti-β5 from R&D systems (AF8035, 1:20 dilution); anti-GRP78 from Genetex (GTX102567, 1:20 dilution); anti Cytochrome c from BD Pharmingen (556432, 1:400 dilution); anti-AIF from Santa Cruz (sc-13116, 1:200 dilution). Antibodies for immunoblotting and immunoprecipitation: anti-ISM came from Biolegend (622201, 1:500 dilution); anti-AIF from Sigma (A7549, 1:100 dilution); anti-cytochrome c from BD Pharmingen (556433, 1:100 dilution); anti-synaptopodin (sc-21537), anti-caspase-8 (sc-5263, 1:200 dilution); anti-caspase-3 (sc-7272, 1:200 dilution); anti-GRP78 (sc-166490, 1:100 dilution) and anti-αvβ5 (sc-13588, 1:100 dilution) were provided by Santacruz.

### 4.2. Animal Models of Proteinuria

All animals were kept in an air-conditioned room with food and tap water ad libitum. All these studies were authorized by an Ethics Committee (#12215). All animal procedures were in accordance with the European Union Guidelines for the care and the use of laboratory animals and were approved by the local ethics committee (Comité National de Reflexion Ethique sur l’expérimentation Animale #05). 

#### 4.2.1. Hypertensive Nephropathy with FSGS Lesions (L-NAME Model)

L-NAME (N^G^-nitro-L-arginine methyl ester) is a competitive inhibitor of NO synthases, that induces high blood pressure, hypoxia, and renal fibrosis [17]. L-NAME (15 mg/kg per day) was administrated in drinking water to male Sprague–Dawley rats (250 g, Envigo. Gannat, France) as previously described [16]. We added NaCl (6 g/L) in the drinking water to accelerate the renal injury as previously detailed [18]. Weekly measurement of proteinuria allowed us to set a threshold of 1 g/mmol of urinary creatinine beyond which severe nephroangiosclerosis lesions were present (mortality rate ~20%) [19]. At this level of urinary protein excretion ratio (UPER), we discontinued L-NAME administration. Then we proceeded to monitor the renal repair at different times after interruption of L-NAME as previously reported [35]. Four groups of animals were studied:-Group W0, sacrificed at the peak of renal damage (UPER > 1 g/mmol) (*n* = 6).-Group W1, animals with UPER >1 g/mmol were sacrificed one week after the interruption of L-NAME (*n* = 5).-Group W5, animals with UPER >1 g/mmol were sacrificed 5 weeks after the interruption of L-NAME (*n* = 4).--CTL group, which did not receive L-NAME and was sacrificed alongside other groups to obtain animals of the same age (*n* = 4).

#### 4.2.2. Puromycin Model

Sprague–Dawley rats were purchased from Envigo (Gannat, France). Puromycine aminonucleoside (PAN) was obtained from Santa Cruz Biotechnology (Dallas, TX, USA). Rats were divided into two groups: PAN-treated and a control group of non-treated rats. Then, eleven 120–150 g male rats from 7 to 8 weeks old were injected subcutaneously with a daily dose of PAN (120 mg/kg) for 10 days. The control untreated group included 7 rats injected with the same volume of saline (0.9% NaCl). Ten days after the last injection, animals were sacrificed and urine, blood, and kidney tissues collected.

#### 4.2.3. Doxorubicin Model

Twelve Sprague–Dawley rats received a single penis-vein injection of a solution of 6 mg/kg doxorubicin (DOXO). As control group, 9 rats received a single injection of saline into the penis-vein. The rats were placed in metabolic cages for urine collection weekly and 30 days after the treatment the rats were sacrificed and urine, blood, and kidney tissues collected for analysis. 

#### 4.2.4. Diabetic Nephropathy Model

This model is used to analyze the role of hyperinsulinemia and obesity in the pathogenesis of non-insulin-dependent diabetes and diabetic nephropathy. C57BL/6J/ob/ob mice were purchased from Jackson Laboratories (Bar Harbor, ME) and maintained in a specific pathogen-free facility with a 12-h light cycle and free access to standard diet and water. Wild type C57BL/6 (*n* = 5) mice were used as control. Mice were killed at 18 weeks of age (*n*= 4), and urine, blood, and kidney tissues were collected. 

#### 4.2.5. Lipopolysaccharide (LPS) Model

Female wild-type BALB/c mice (8–10 weeks old) were purchased from Charles River Laboratories (Lyon, France), subsequently housed and given free access to food and water. After two weeks, 8 mice were injected with LPS i.p. (10 µg/g of weight; Sigma Aldrich, St. Louis, MI, USA). A control group of 5 mice received an equal volume of saline (i.p.). Blood, urine, and kidney tissues were collected 24 h after LPS or saline treatment. For all these models, blood and urines samples were immediately spun for 10 min at 4000 rpm and frozen at −80 °C. The subsequent biochemical measurements (albuminuria, creatininuria) were realized on the Architect system (Abbott Biagnostic) by the Department of Biochimie–Hormonologie, Tenon Hospital (Paris, France).

### 4.3. DNA Microarrays Analysis

The hybridization protocol and the computer analysis were carried out in collaboration with the genomics platform of the Cochin Institute (Genom’IC). One quarter of each animal’s kidney was thawed and ground, and the RNA was extracted using Trizol (Invitrogen, Camarillo, CA, USA). The quantity, purity, and quality of the RNAs were verified using Nanodrop (Thermo Fisher Scientific, Waltham, MA., USA) and a Bioanalyzer (Agilent Technologies, Santa Clara, CA, USA).

The RNAs of interest were diluted to a concentration of 100 ng/μL before starting the synthesis of the first strands of cDNA. The cDNAs were synthesized using a NuGEN kit (San Carlos, CA, USA), the second cDNA strands from the previous ones; the double-stranded cDNA thus obtained was purified by the magnetic method. Linear amplification (SPIA technology) of these double-stranded DNAs was then performed to obtain a sufficient amount of hybridization material on the chips while maintaining the respective proportions of each RNA of the original sample. After purification, the amplified cDNAs were enzymatically fragmented and associated with a fluorescent label. Finally, the samples were denatured at 99 °C and hybridized on the chip (Affymetrix, Santa Clara, CA, USA) for 17 h at 45 °C. At the end of the hybridization, the chips were washed and placed in a high-resolution scanner in order to obtain the expression values corresponding to each probe. The fluorescence values obtained for each of the probes were normalized using the RMA algorithm (Robust Microarray Analysis) [36]. This approach allows reduction of differences in hybridization between the chips and to facilitate the identification of true differences of expression. The list of transcripts differentially expressed in the three groups compared to the control group was then imported in the software IPA (Ingenuity Systems) for subsequent pathway analysis. The raw data were deposited in the gene expression omnibus (GEO) database of NCBI with the accession number GSE151690.

### 4.4. Immunohistochemistry Analysis 

Kidneys collected from animals were snap-frozen in liquid nitrogen and fixed in acetone. For tissue staining, frozen cryostat sections (4 μm) were treated by BSA 3% in PBS over 30 min, incubated with ISM-1 antibody, and detected by secondary antibodies purchased from Nichirei-Histofine Simple Stain rabbit MAX PO (Tokyo, Japan). Revelation was achieved with 3-amino-9-ethylcarbazole AEC (Dako, Carpinteria, CA, USA). For examination in fluorescent conditions, Alexa Flour Plus secondary antibodies were used (ThermoFisher, Illkirch-Graffenstaden, France). 

For transmission electron microscopy (TEM) analysis, kidneys were cut into small pieces and immersed in 2.5% glutaraldehyde containing 1% tannic acid in 0.1 M PBS for 2 h at 4 °C. Samples were post fixed with 1% OsO4, dehydrated and embedded in epoxy resin. Ultrathin sections were stained with uranyl acetate and lead citrate and then examined under a Philips CM10 electron microscope. Ultrathin frozen sections were processed for indirect immunogold labeling, as described [37].

### 4.5. Isolation of Glomeruli and qPCR Analysis

Decapsulated glomeruli were isolated as described previously [38]. Briefly, freshly isolated renal cortex was mixed and digested by collagenase I (2 mg/mL; Gibco, Billings, MT, USA) in RPMI 1640 (Life Technologies, Carlsbad, CA, USA) for 2 min at 37 °C, then collagenase I was inactivated with RPMI 1640 + 10% FCS (Abcys, Paris, France). Tissues were then passed through a 100-µm cell strainer and 40-μm cell strainer for rat tissues and through 70-μm cell strainer and 40-μm cell strainer (BD falcon) for mouse’s tissues in PBS (Life Technologies) + 0.5% BSA (Euromedex, Souffelweyersheim, France). Glomeruli, adhering to the 40-μm cell strainer, were taken from the cell strainer with PBS + 0.5% BSA injected under pressure, then washed twice in PBS. Isolated glomeruli were then resuspended in RLT extraction buffer (Qiagen, Venlo, The Netherlands) and frozen at −80 °C until total RNA extraction. The glomerular mRNA was extracted by using the EZ-10 Spin Column Total RNA Kit (BD Life Science) according to the manufacturer’s instructions. RNA concentration was measured by using the NanoDrop1000 spectrophotometer (Thermo Fischer Scientific Biosciences, Germany). RNA were reverse transcribed using the Maxima First Strand cDNA Synthesis Kit (Thermo Fischer Scientific Biosciences, Germany) and PCR was performed using SYBR green (Roche Diagnostics, Meylan, France) and specific primers (Appendix A) purchased from Euronfins Scientific (Paris, France) on a Light Cycler 480 (Roche Diagnostics, Meylan, France). Expression levels were normalized to the house keeping gene, HPRT (hypoxanthine-guanine phosphoribosyl transferase) or GusB (beta-glucuronidase) using the Light Cycler^®^ advanced relative quantification program (Roche). 

### 4.6. Podocytes Cell Culture

Conditionally immortalized transgenic human podocytes enabled via a thermosensitive variant of SV-40 were provided by Dr. Moin Saleem [39]. Briefly, podocytes were cultured in RPMI-1640 with ITS (insulin (10 μg/mL), transferrin (5.5 μg/mL), selenium (5 ng/mL Na selenite)), heat-inactivated fetal bovine serum (10 % *v*/*v*), 100 U/mL penicillin, and 0.1 mg/mL streptomycin (Invitrogen, Carlsbad, CA, USA) at 33 °C under a humidified atmosphere of 95 % air and 5% CO_2_ with change of the medium every 2 days. Under these conditions of culture, podocytes exhibit a proliferative phenotype (permissive conditions). Podocytes at 50–60% of confluence were thermo-switched from 33 °C to 37 °C for differentiation under non-permissive conditions (same medium without ITS) for 14 days. Differentiated podocytes were used in all experiments.

### 4.7. Cell Viability and TUNEL Assay

The viability of cultured podocytes was assessed using the MTT assay. The tetrazolium salt 3-(4,5-dimethyl-2-thiazolyl)-2,5-diphenyl-2H-tetrazolium bromide (MTT) assay was purchased from Sigma-Aldrich (M5655). Cells were cultured in 96-well plates (20,000 cells/well) and incubated in 200 μL of non-permissive medium (37 °C) in different conditions (ISM1, inhibitors). After incubation, MTT was added to the medium (10 μg/mL), and the plates incubated at 37 °C for 2 h and the MTT solution removed. The MTT-formazan produced was solubilized with dimethyl sulfoxide (DMSO) and measured at 590 nm in a microplate reader (Thermo Fisher).

Apoptosis was measured using a TUNEL assay kit obtained (from Abcam, San Francisco, CA, USA). Podocytes were fixed for 15 min using 1% paraformaldehyde and after fixation, the cells were rinsed with PBS, treated with 70% ethanol, and incubated for 30 min on ice. After incubation, the cells were washed with the provided buffer and 50 µL of DNA labeling solution was added to the cells, for 1 h at 37 °C. The cells were then washed with buffer and re-suspended in DAPI solution for 5 min and incubated in the dark. The cells were then observed under a Leica DM-RXA23D microscope (Wetzlar, Germany).

### 4.8. Immunofluorescence 

Podocytes were seeded onto uncoated coverslips (Marienfeld GmbH, Germany) and incubated with recombinant ISM1 under different conditions. After washing in PBS, cells were fixed in paraformaldehyde (3%) for 15 min, washed with PBS, and incubated for 5 min in 100 mM NH4Cl in PBS. Cells were permeabilized with 0.1% saponin, saturated with fetal bovine serum 3% in PBS and incubated for 1 h with primary antibody. After washing, cells were incubated with an appropriated secondary antibody conjugated with Alexa Fluor 594 or 488, (Invitrogen, Waltham, MA, USA). Cells were mounted in Mowiol medium and fluorescence observed in a Leica DM-RXA23D microscope (Wetzlar, Germany). 

### 4.9. Fluorimetric Determination of Mitochondrial Membrane Potential (ΔΨm)

The measurement of the mitochondrial membrane potential (ΔΨm) was assessed by using the lipophilic cationic dye JC-1 (5,5,6,6′-tetrachloro-1,1′,3,3′-tetra-ethylbenzimidazolocarbo-cyanine iodide; Sigma) which is accumulated within the mitochondria. When excited at 490 nm, monomers of JC-1 fluoresce at 530 nm. Higher concentrations of JC-1 form aggregates which emit maximally at 590 nm. Consequently, cells with high ΔΨm fluoresce orange, while cells with low ΔΨm fluoresce green. Loss of ΔΨm is considered as a marker of the onset of apoptosis. Podocytes were seeded into 96-well plates or coverslips at a density of 20000 cells/well in different conditions. Then JC-1 (20 µM) was added to growth medium and incubated 30 min at 37 °C. The fluorescence in each well was measured in a fluoroskan^®^ (Thermo Fisher) microplate fluorometer (excitation, 490 nm; emission, 530 nm for JC-1 monomer and 590 nm for JC-1 aggregates). The mitochondrial membrane potential (ΔΨm) from each group was calculated as the fluorescence ratio of red to green and expressed as a percentage of the control.

### 4.10. Western Blotting and Immunoprecipitation Assays

Cultured podocytes were lysed in RIPA buffer (Thermo Fisher), containing a protease inhibitor cocktail (Thermo Fisher). Equal amounts of protein (25–50 μg), were separated by SDS-PAGE, transferred onto a PVDF membrane, and blocked (3% BSA in TBS-Tween). Membranes were subsequently probed with the indicated primary antibodies. Appropriate HRP-conjugated secondary antibodies were used, and the signal was detected with an Immobilon Western kit (Millipore, Molsheim, France). Densitometric analysis of visualized bands was performed using Image J software.

For immunoprecipitation experiments, podocytes were lysed in IP lysis buffer (Thermo Fisher). Equal amounts of protein were immunoprecipitated with saturated amounts of anti-GRP78 or anti-αvβ5 antibodies overnight at 4 °C, followed by incubation with 20 µL of protein G-Sepharose beads for 1 h at 4 °C. The beads were then washed, boiled, and the supernatants were used to immunoblot using an anti-ISM antibody. All blots were performed in three independent experiments.

### 4.11. Statistical Analysis

All in vitro experiments were repeated at least 3 times. Values are expressed as the mean ± SEM. Student’s *t* test was used to observe differences between two groups. ANOVA test following by t test was used when more than two groups were present. All statistical analyses were performed using GraphPad Prism 5.0. Differences were considered statistically significant when the *p* value was <0.05.

### 4.12. In Silico Analysis of ISM1-Related Pathways of the L-NAME Model

Transcriptome analysis was performed in R software environment 4.1.0. Expression heatmaps were drawn with pheatmap R-package version 1.0.12. Alluvial plots were drawn with ggalluvial P-package version 0.12.3. ISM1 supervised expression profile was determined with the Pavlidis Template Matching algorithm [40] during the time course experiment of the GEO dataset GSE151609. Functional enrichment was performed with with Toopgene web suite application [41]. Functional enrichment networks were drawn with Cytoscape standalone application version 3.6.0 [42]. Text mining related gene-function associations were performed with GeneValorization [43] application on NCBI database.

## 5. Conclusions

In the present work, we describe for the first time to our knowledge a pathophysiological role of ISM1 in the kidney. This interest in ISM1 emerged because of the variability of its expression according to disease progression. This provided a first indication that ISM1 was associated with renal dysfunction. Furthermore, our results showed that ISM1 was overexpressed in five different models of glomerular disease. This abnormal expression occurs specifically in podocytes where it produces apoptosis and glomerular damage.

## Figures and Tables

**Figure 1 ijms-24-02723-f001:**
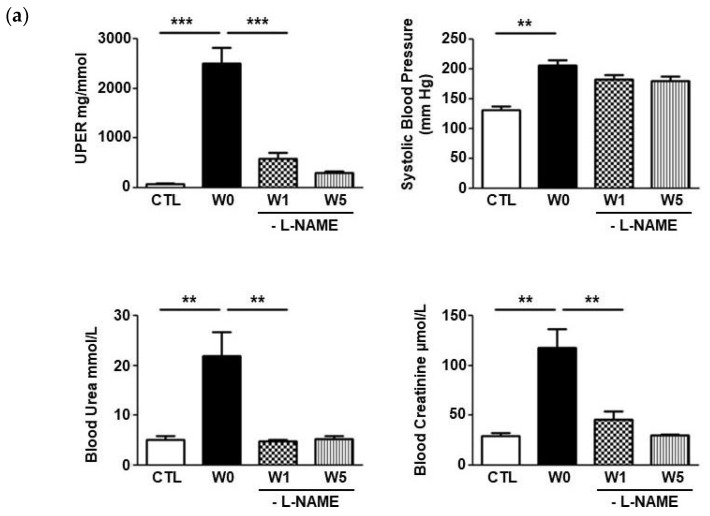
**L-NAME impaired the renal function in rats and induced renal injuries.** (**a**) *Blood pressure and renal function in L-NAME rat model*. Parameters studied included UPER, uremia and creatinemia in CTL, W0, W1, and W5 groups. At the peak of the disease (W0), the rats developed an elevated UPER accompanied by high systolic blood pressure and high levels of urea and serum creatinine. After stopping L-NAME treatment (W1 and W5), renal function recovered to normal (*** *p* < 0.001 and ** *p* < 0.01). (**b**) *Histological evaluation of renal tissues.* Representative examples of renal samples stained with Masson’s trichrome stain in Control (CTL) and L-NAME-treated rats. Glomerulosclerosis (black arrow) and intertial fibrosis (asterix) on the right photo. Bar = 10 µm.

**Figure 2 ijms-24-02723-f002:**
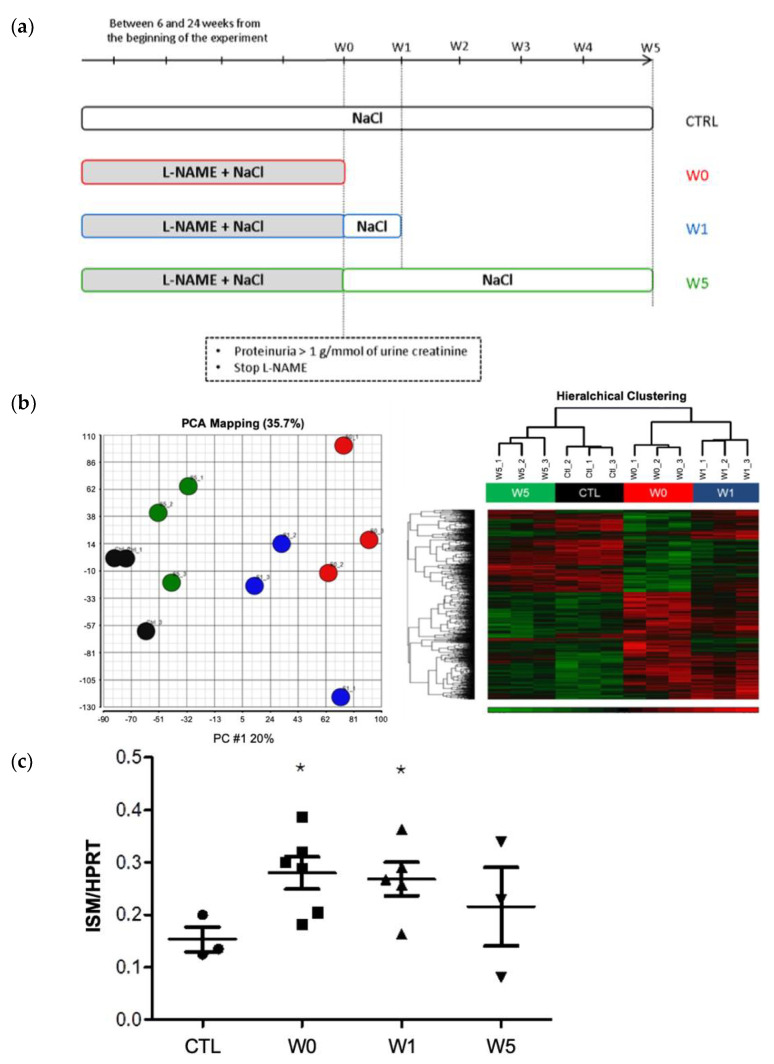
**ISM1 gene expression in L-NAME model.** (**a**) Experimental protocol of L-NAME model. When urinary protein excretion ratio was over 1 g/mmol of creatinine, after 6 to 24 weeks, NG-nitro-L-arginine methyl ester (L-NAME) was removed. (**b**) *Left panel.* Segregation of transcriptomic data of each animal by principal component analysis: Each animal is represented by a circle in the corresponding group (CTL in black, W0 in red, W1 in blue, and W5 in green). *Right panel.* Hierarchical grouping of transcriptomic data from each animal: Each line corresponds to a gene and each column to an animal. The red color indicates an overexpression and the green color an under expression. This representation clearly shows a variation of gene expression from one group to another. (**c**) ISM1 is overexpressed in glomeruli of the L-NAME rat model. mRNA expression of ISM1 from kidney of CTL rats, L-NAME treated rats (W0), 1 week L-NAME removal rats (W1), and 5 weeks L-NAME removal rats (W5). mRNA of ISM1 is significantly overexpressed in the kidney of W0 and W1 L-NAME rats (* *p* < 0.05 vs control group).

**Figure 3 ijms-24-02723-f003:**
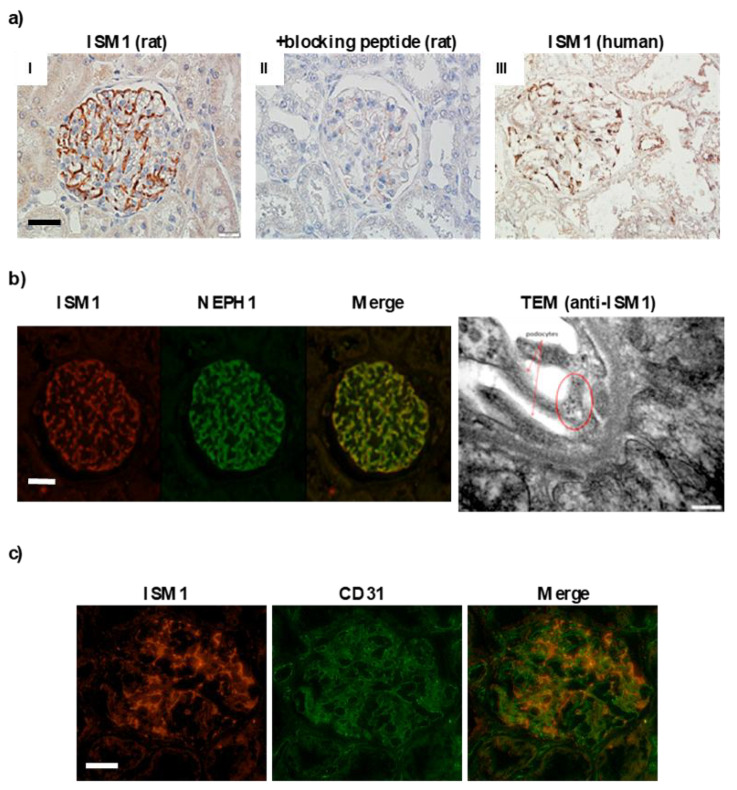
**Physiological expression of ISM1 in rodents and human kidneys.** (**a**) Glomerular expression of ISM1 in kidney from rats (**I**,**II**) and human (**III**). ISM1 is expressed exclusively in podocytes. (**b**) ISM1 expression is localized in podocytes. Immunofluorescence showing colocalization between ISM1 and nephrin. Immunogold electron microscopy: ISM1 is localized in the cytoplasm (circled in red) and in the foot process of podocytes (red arrow). (**c**) ISM1 colocalized poorly with the endothelial cell marker CD31. Bar = 10 µm and 400 nm for TEM.

**Figure 4 ijms-24-02723-f004:**
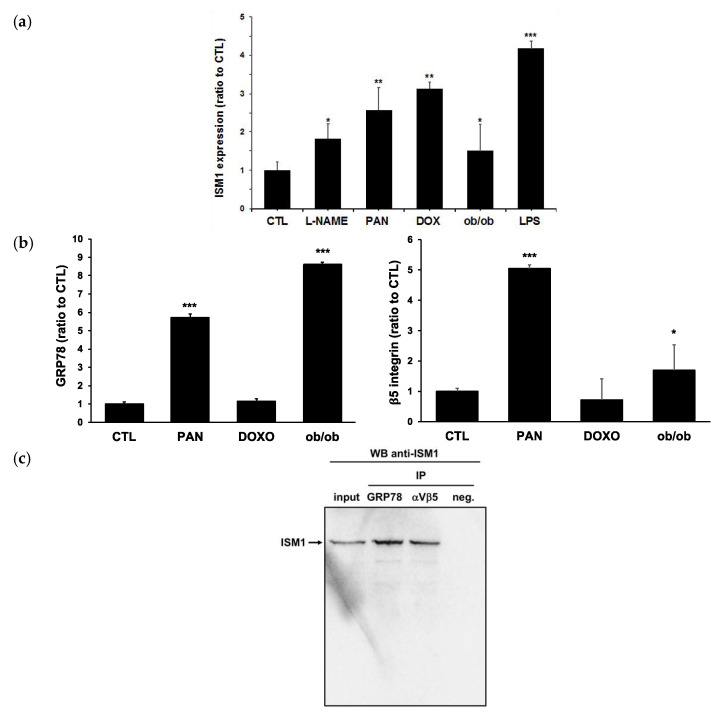
**Expression of ISM1 and its receptors in different models of glomerulopathies.** Estimation by RT-qPCR (*n* = 6): (**a**) ISM1 expression in different models of proteinuria (L-NAME (W0), doxorubicin, puromycin, obesity, and LPS. Values normalized to CTL; (**b**) Expression of GRP78 (left) and integrin-β5 (right) in doxorubicin, puromycin, and obesity models. *, *p* < 0.05 vs CTL **, *p* < 0.01 vs CTL ***, *p* < 0.001 vs CTL (**c**) Interaction between ISM1 and its GRP78 and integrin αvβ5 receptors in cultured podocytes. Immunoprecipitation (IP) of GRP78 and integrin αvβ5 using podocyte cells and then, Western blot of ISM1. Negative control with unspecific antibody. The ISM1 interacts with its receptors in podocytes, in basal conditions, (*n* = 2).

**Figure 5 ijms-24-02723-f005:**
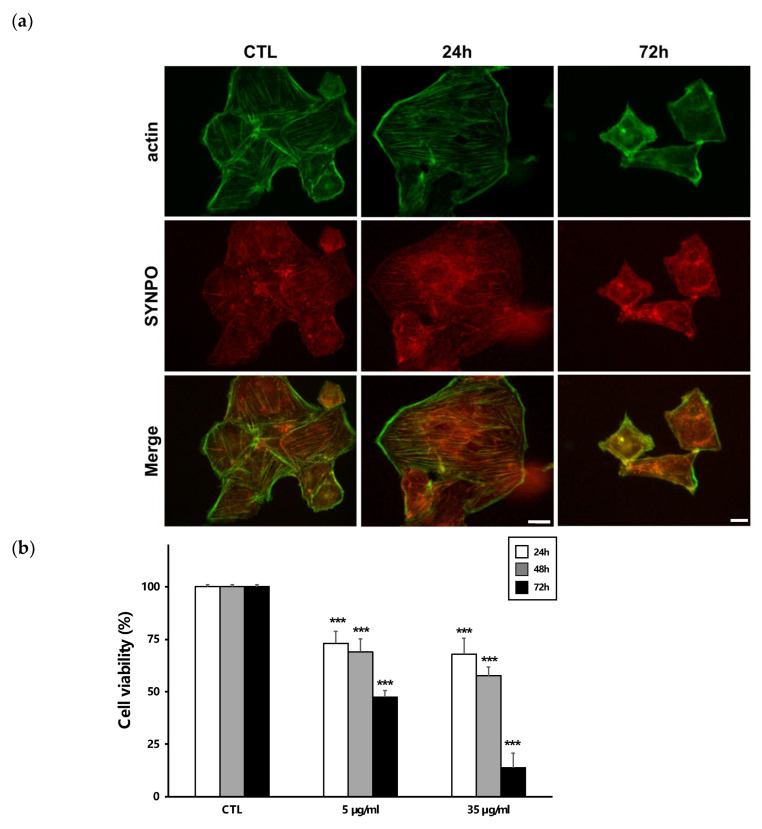
**Recombinant ISM1 decreased podocyte viability**. (**a**) Staining with phalloidin-FITC showed that the F-actin cytoskeleton was disorganized after strong stimuli with ISM1 in podocytes. Synaptopodin (SYNPO) lost its co-localization with actin upon addition of high concentrations of ISM1 to podocytes. (**b**) Podocytes were treated with recombinant ISM1 (5 and 35 µg/mL) for 24, 48, and 72 h. ISM1 significantly decreased the podocyte viability. ***, *p* < 0.001 vs CTL. (**c**,**d**) Podocytes were treated with ISM1 (5 and 35 µg/mL) for 24 and 72 h in the presence or absence of QVD-oph, a pan-caspase inhibitor. QVD-oph prevented ISM1-induced decrease in viability under mild conditions (5 µg/mL, 24 h) (**c**). At higher doses or longer incubations of ISM1, QVD-oph only partially inhibited podocyte mortality (**d**). **, *p* < 0.01 vs. CTL ***, *p* < 0.001 vs CTL; ##, *p* < 0.01 treated vs untreated QVD, #, *p* < 0.05 treated vs untreated QVD. Bars = 20 µm for CTR and 24h and 50 µm for 72h.

**Figure 6 ijms-24-02723-f006:**
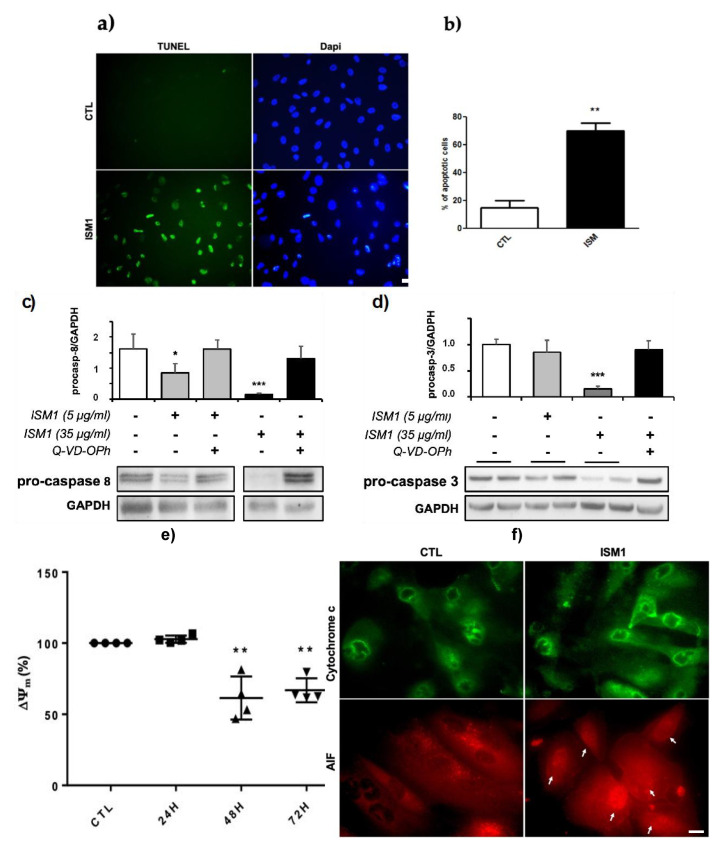
**ISM1 induces apoptosis in podocytes.** (**a**) Estimation of ISM1-induced apoptosis by TUNEL assay after un strong stimulus ISM1 (35 µg/mL) for 72 h. Cells were observed under fluorescence microscopy. (**b**) Apopotic cells were quantified. ISM1 induced an emergence of TUNEL positive cells corresponding to apoptotic cells (** *p* < 0.01 vs control group; *n* = 3). (**c**,**d**) Evidence of caspases activation. Representative immunoblots with quantitative data of densitometry are shown in (**c**) for caspase 8 and in (**d**) for caspase 3. ISM1induced the degradation of pro-caspases 8 and 3 (** *p*< 0.01; * *p* < 0.05 vs control group; *n* = 4 blots) this was inhibited by QVD-OPh. (**e**) ISM1 decreases the mitochondrial membrane potential of podocytes (ΔΨm). Podocytes were treated with ISM1 (35 µg/mL) and quantification of ΔΨm was assessed by the JC-1 probe. ISM1 decreased significantly the ΔΨm of podocytes from 48 h to 72 h (*** *p* < 0.01; ** *p* < 0.01 vs control group). (**f**) AIF was released from mitochondria to the cytosol and the nucleus after ISM1 treatment (arrows). In contrast, cytochrome c remained confined in mitochondria after ISM1 treatment. Bars = 50 µm for (**a**) and 20 µm for (**f**).

**Figure 7 ijms-24-02723-f007:**
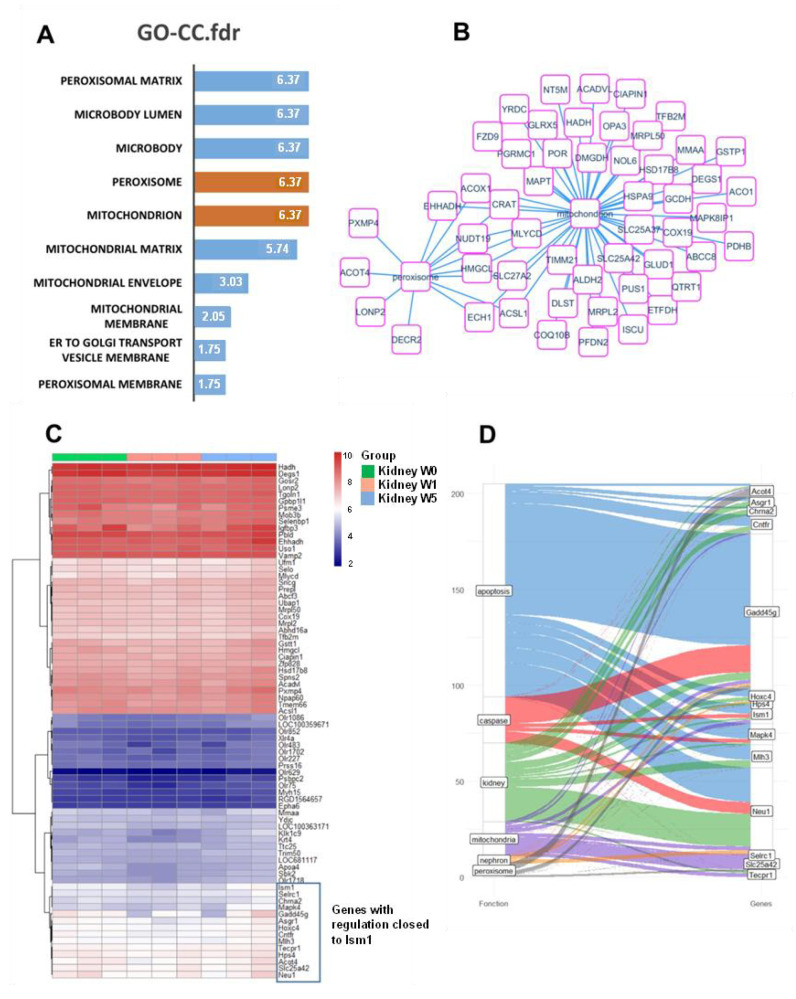
**Mitochondria and peroxisome regulation during ISM1 time course model during kidney regeneration.** (**A**) Functional enrichment on gene ontology cellular component database of genes found correlated to ISM1 during kidney regeneration (fdr: false discovery rate): barplot of negative log10 FDR adjust *p*-values; (**B**) Functional enrichment network performed with mitochondria and peroxisome related genes found correlated to ISM1; (**C**) Expression heatmap of best 75 genes found correlated to ISM1 expression kidney regeneration: blue gene cluster defined genes with regulation closed to ISM1; (**D**) Alluvial plot built on gene-function found in NCBI database with genes highlighted closely regulated with ISM1 during kidney regeneration (quantitative colored links represent the numbers of NCBI-Pubmed articles connected).

**Figure 8 ijms-24-02723-f008:**
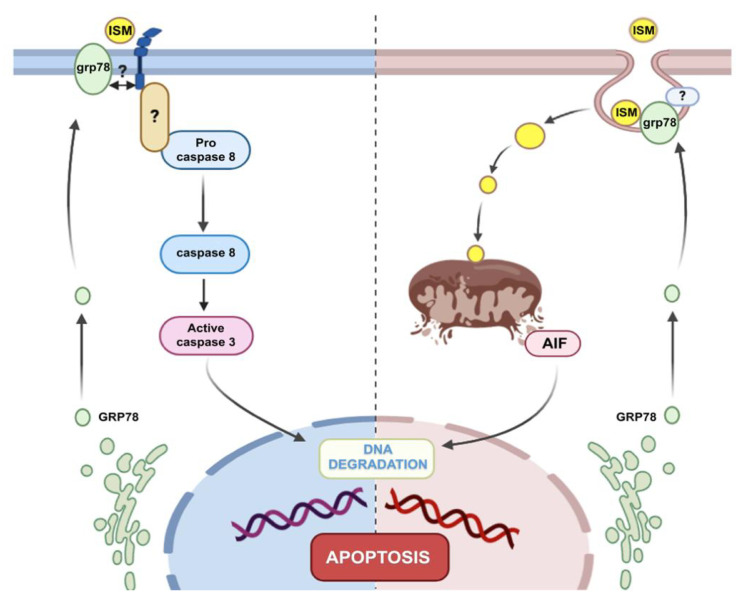
**Proposed mechanism of the deleterious action of ISM1 on podocytes.** (Left) Caspase-dependent pathway. At low concentration, extracellular ISM1might bind αvβ5 and/or GRP78 at the cell surface, triggering activation of procaspase-8 and the downstream effector caspase-3, which would activate a caspase-dependent DNAse (CAD) to produce DNA fragmentation (left). At higher concentrations, ISM1 is internalized by endosomal trafficking and would induce the activation of a caspase-independent proapoptotic pathway. Thus, ISM1 potentially interacts with GRP78 on the cell surface and would trigger its endocytic transport into the mitochondria. Subsequently, ISM1 triggers the permeabilization of the outer mitochondrial membrane and the specific release of AIF into the cytosol. Ultimately, AIF translocates to the nucleus where it contributes to DNA fragmentation and chromatin condensation.

## Data Availability

Data is contained within the article or Appendix A.

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
