# Peer review of "The Angiogenesis Inhibitor Isthmin-1 (ISM1) Is Overexpressed in Experimental Models of Glomerulopathy and Impairs the Viability of Podocytes"

_ijms, 2023, doi:10.3390/ijms24032723_

Round 1

Reviewer 1 Report

line 94: please describe the abbreviation UPER, where it is first used. Most will understand urine protein excretion rate but needs to be clarified 

Author Response

Dear Editor and reviewers,

Thank you very much for your comments and suggestions.

We changed and modified the figure and text as proposed by reviewers. Modifications are in red in the new version of the manuscript.

Reviewer 1:

Line 94: please describe the abbreviation UPER, where it is first used. Most will understand urine protein excretion rate but needs to be clarified.

We described the abbreviation of UPER as requested.

We hope that these modifications will made the article suitable for publication in IJMS.

Best regards

JJ Boffa

Reviewer 2 Report

In this manuscript by Virgilia et al., the authors systematically characterized the effects of Isthmin-1 on the progression of FSGS via regulating the viability of podocytes via in vivo and ex vivo experiments, and report that during the progression of FSGS, ISM1 is overexpressed, which promotes podocyte apoptosis in both caspase-in/dependent ways. Overall, the manuscript is very well written. The experimental approach is straightforward and clearly described. However, there are a few points that require additional investigation to strengthen the hypothesis. For this reason, I have some suggestions to improve the manuscript:

Major:

Recombinant ISM1 induced podocyte injury in vitro, can this ISM1 toxicity can be repeated in animal models? If the blocking peptide can attenuate this toxicity in vivo?

Minor:

1. In figure 1a, please clarify the Blood pressure, which is Systolic blood pressure or mean arterial pressure. Additionally, the Uremia and creatininemia graphs, the y axis title uremia and creatiniemia are incorrect. Uremia and creatininemia describes phenomena of high levels in the blood. Here, I think they should be  blood urea and blood creatinine, respectively.

2. In figure 1b, please label and explain the changes (arteriolosclerosis, interstitial fibrosis, and tubular dilation) in the graph.

3. Please provide scale bar for all histological images.

Author Response

Dear Editor and reviewers,

Thank you very much for your comments and suggestions.

We changed and modified the figure and text as proposed by reviewers. Modifications are in red in the new version of the manuscript.

Reviewer 2:

Major: Recombinant ISM1 induced podocyte injury in vitro, can this ISM1 toxicity can be repeated in animal models? If the blocking peptide can attenuate this toxicity in vivo?

We have tested different approaches to verify the ISM1 toxicity in vivo. First, we injected recombinant ISM (MBS1313413 from Mybiosource) at the dose of 300 nM in anesthetized animals and measured UPER for several hours. Secondly, we generated a cDNA construction of murine ISM1 and we used an Adeno-associated virus (AAV) to harbor it. Mice were IM injected with this AAV to induce increase expression of ISM1 in vivo. Unfortunately, results obtained by both methods were not conclusive. Currently, we create a conditional overexpression of ISM1 using a ROSA knock-in mice to answer this important issue.

To date, to our knowledge, no blocking peptide capable to inhibit the interaction of ISM1 and its receptors (GRP78 and avb5 integrin) has been described. The specific sites of interaction are not known. The blocking peptide used in our experiment concerned the interaction of ISM1 with a specific antibody.

Minor:

  1. In figure 1a, please clarify the Blood pressure, which is Systolic blood pressure or mean arterial pressure. Additionally, the Uremia and creatininemia graphs, the y axis title uremia and creatininemia are incorrect. Uremia and creatininemia describes phenomena of high levels in the blood. Here, I think they should be blood urea and blood creatinine, respectively.
  2. In figure 1b, please label and explain the changes (arteriolosclerosis, interstitial fibrosis, and tubular dilation) in the graph.
  3. Please provide scale bar for all histological images.

We agree with all these suggestions and we thank to the referee for that. We have included all those suggestions in the manuscript.

We hope that these modifications will made the article suitable for publication in IJMS.

Best regards

JJ Boffa
